

# Surviving on the edge: present and future effects of climate warming on the common frog (*Rana temporaria*) population in the Montseny massif (NE Iberia)

Albert Montori[1] and Fèlix Amat[2]

[1] Herpetology, Centre de Recerca i Estudis Ambientals de Calafell (CREAC/GRENP), Calafell, Catalonia, Spain
[2] Herpetological Section (BiBIO), Natural History Museum of Granollers, Granollers, Catalonia, Spain

## ABSTRACT

The Montseny massif shelters the southernmost western populations of common frogs (*Rana temporaria*) that live in a Mediterranean climate, one which poses a challenge for the species' persistence in a scenario of rising temperatures. We evaluated the effect of climate change at three levels. First, we analysed if there has been an advancement in the onset of spawning period due to the increase in temperatures. Second, we analysed the impact of climatic variables on the onset of the spawning period and, third, how the distribution of this species could vary according to the predictions with regard to rising temperatures for the end of this century. From 2009 to 2021, we found there had been an increase in temperatures of 0.439 °C/decade, more than the 0.1 °C indicated by estimates for the second half of the previous century. We found an advancement in the onset of the reproduction process of 26 days/decade for the period 2009–2022, a change that has been even more marked during the last eight years, when data were annually recorded. Minimum temperatures and the absence of frost days in the week prior to the onset of the spawning period determine the start of reproduction. Predictions on habitat availability for spawning provided by climatic niche analysis for the period 2021–2100 show a potential contraction of the species range in the Montseny and, remarkably, much isolation from the neighbouring populations.

# INTRODUCTION

Climate change due to increases in human greenhouse gas emissions since the beginning of the industrial revolution up to the present day is now considered one of the major threats to biodiversity in the twenty-first century and one which underlies several reported extinction events (*Thomas et al., 2004*; *Parmesan, 2006*; *Thuiller et al., 2011*). Climate is rapidly changing and the same trend that points towards increases in global temperatures has been forecast for the coming decades (*IPCC, 2021*; *Seneviratne et al., 2014*). Understanding how species respond to increases in temperatures and changes in

Corresponding author
Albert Montori,
amontori@gmail.com

precipitation, both at a population and species level, is one of the most important questions to be answered. With regard to amphibians, *Sheridan et al. (2018)* indicate that changes in body size and breeding phenology are the two major ecological consequences of climate change in the wood frog (*Lithobates sylvaticus*), but it is unclear how rising temperatures interact with these variables. *Reading (2007)* reported a reduction of survival, body condition, size, and egg laying in the common toad (*Bufo bufo*), and *Blaustein et al. (2010)*, reported changes in developmental rates of eggs and larvae.

Changes in temperature or precipitation can potentially influence the timing of amphibian reproduction (*Blaustein et al., 2010*). An earlier onset of spring breeding behaviour correlated with a warming climate has been observed among various pond-breeding frogs and toads in Europe (*Beebee, 1995*; *Tryjanowski, Mariusz & Sparks, 2003*; *Scott, Pithart & Adamson, 2008*) and Asia (*Kusano & Inoue, 2008*; *Primack et al., 2009*). However, other species may exhibit either a trend towards a later onset of spring breeding or no trend at all (*Todd et al., 2010*; *Klaus & Lougheed, 2013*). *While & Uller (2014)* and *Ficetola & Maiorano (2016)* concluded that this group is strongly influenced by climate change after carrying out a meta-analysis with worldwide phenological data of amphibians. However, they didn't find a clear pattern for the impact climate can have on amphibian population parameters such as abundance, survival, breeding success and morphology. Both authors agreed on the low importance of rainfall changes, the effects of climate change on offspring and the strong relationship between the advancement in the onset of spawning and rising temperatures. However, in some cases, an increase in precipitation could enhance phenological advancement (*Timm, McGarigal & Compton, 2007*; *Todd et al., 2010*; *Green, 2017*), especially in warm and dry areas where precipitation is a limiting factor (*Ficetola & Maiorano, 2016*). Shifts in phenology due to climate change have been observed by many authors in several species (*Beebee, 1995*; *Parmesan, 2006*; *Walther et al., 2002*; *While & Uller, 2014*; *Ficetola & Maiorano, 2016*), with evidence of earlier breeding with a shifting climate and *Chadwick, Slater & Ormerod (2006)* showed trends of earlier breeding associated with shifts in the climate in *Triturus* species. However, findings from several independent studies reveal that the impacts of a changing climate on breeding times may be more complex. With regard to Common Frog, *Rana temporaria*, the most recent studies found a close relationship between rising temperatures and an advanced reproduction date (*While & Uller, 2014*; *Ficetola & Maiorano, 2016*; *Bison et al. 2021*).

One of the research areas pertaining to the effects of climate change on species entails measuring the expected shift in the ranges they will experience in the future. Changes in geographic distribution can occur because climate change affects population growth and their survival or the ability to colonise new areas (*McCaffery & Maxell, 2010*), mediated by the species' physiological adaptation to a range of environmental conditions (*Greenberg & Palen, 2021*). Modelling habitat suitability across time using current data is commonly used to predict how amphibian ranges will change, an event shaped by the effect of climate change (see for example *García & Ortega-Huerta, 2013*; *Duan et al., 2016*; *Agudelo, Urbina-Cardona & Armenteras-Pascual, 2019*; *Kim et al., 2021*). However, the increase in data availability on species distribution gathered over recent decades, has recently shown
shifts in the species ranges. For example, *Enriquez-Urzelai et al. (2019a)* demonstrated that most Iberian amphibians have changed their altitudinal and latitudinal distributions, moving up to the north or to higher altitudes in the last 20–25 years as a response to rising temperatures.

The common frog is widely found across the temperate and cold regions in Europe, from the sea level up to 2,700 m a.s.l. reaching a latitude of 70° North, above the Arctic Polar Circle. The southernmost populations of the species are located in the Rodopi mountain range in Greece, in Eastern Europe (*Kuzmin et al., 2009*). The second most southern rim in which the common frog is found in Western Europe, is the Montseny massif, that is the southernmost foothill of the Transversal Catalan Range placed in Northeast Iberia (Fig. 1). Common frogs breed in shallow, still, fresh water such as ponds, lakes or marshes. The onset of spawning will vary depending on the altitude and latitude, occurs between February to late June, but generally in March-April over the main part of the range. In the Atlantic region of Iberian Peninsula, the spawning occurs earlier, in November-December, due to the abundant rain and mild temperatures (*Bea, Rodríguez-Teijeiro & Jover, 1986*). Outside the breeding season, common frogs live a solitary life in damp places near ponds or marshes or in long grass.

Although the climate in the Montseny is typically Mediterranean, the massif's altitude and the northern and eastern exposure of the main hillsides provide humid and temperate conditions (*Panareda-Clopés, 1979*), thus allowing the persistence of populations of this cold-adapted species. Thus, this massif harbours rear edge populations of the common frog, where the species experiences a small temperature buffer in comparison with most of the range, making these populations a good model to study the effects of climate change on temperate amphibians (*Hampe & Petit, 2005*; *Habibzadeh et al., 2021*).

Furthermore, the models of climatic change (*González-Hidalgo et al., 2009*; *Peñuelas et al., 2021*) for the Mediterranean area forecast more irregularity in precipitation patterns, specifically more rainfall in autumn than in spring, and concentrated over a few days in some years, while in others, there will be heavy droughts. Air temperature is rising (*Vicente-Serrano et al., 2014*) and in the Montseny the variation in the average air temperature in Turó de l'Home (1,712 m a.s.l.) for the period 1950–1999 was 0.3 °C/decade (*Peñuelas & Boada, 2003*), and *Minuartia Estudis Ambentals (2016)*, shows a rate of increase with a similar trend (+0.24 °C/decade), for the period 1950–2014, with a higher average seasonal increase in spring and a lower one in autumn.

Our objectives were to understand how the environmental variables influence the onset of breeding phenology of the common frog in recent years, and to explore the changes that this species faces in the context of climate change in the edge of distribution in the Montseny massif using niche modelling. Specifically, we examined the following issues: (i) we analysed whether there has been an advancement in the onset of the breeding period in recent years as a consequence of rises in global temperatures; (ii) how and which meteorological variables influence this onset, and (iii) whether the species will experience, in the Montseny Massif, a contraction of its distribution range in the 2,100 scenario as a consequence of global warming.
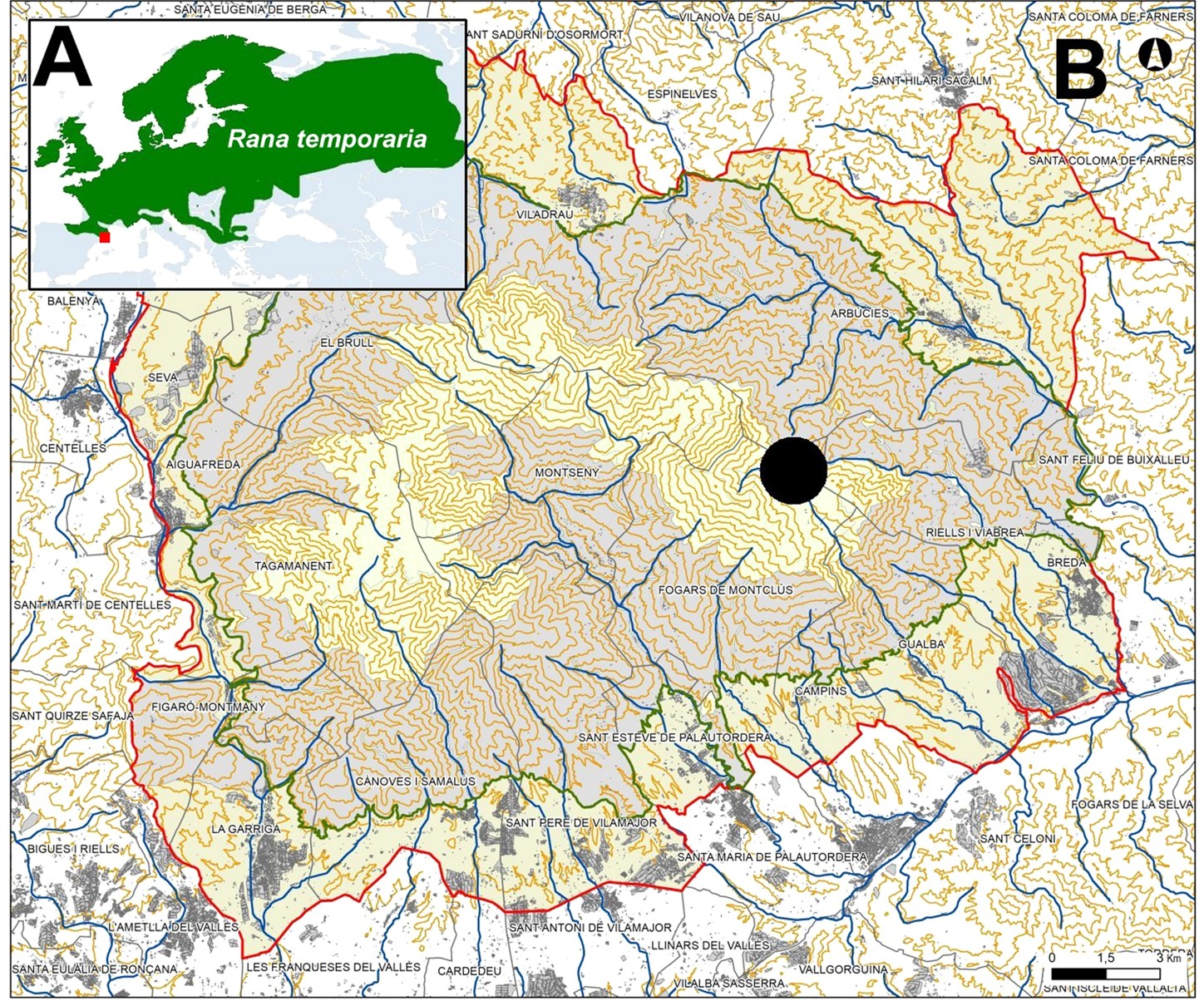

**Figure 1 Geographic range of *Rana temporaria* in the Palearctic and in northeaster Iberia, and location of the study site in the Montseny Massif (black circle and red square, respectively).** (A) Map of global distribution of the common frog. (B) Map of the Montseny massif. Green line: Administrative limit of Natural Park of Montseny. Red Line: Administrative limit of Natural Park and Biosphere Reserve of Montseny.

## MATERIALS AND METHODS

### Study area

A phenological surveis were conducted in the Santa Fe Valley situated on the eastern side of the Montseny massif (UTM31N 5524) at 1,100 m a.s.l (Fig. 1). At the study site, the monthly mean temperature ranges from 3 °C in January to 20 °C in August and the mean annual precipitation is approximately 1,000 mm, occurring mostly as rain in autumn and spring, with occasional snow in winter. The valley is primarily forested with beech (*Fagus*

*sylvatica*) and divided by streams that produce many wetlands and ponds. We extended the study area to all the populations in the Montseny massif to gather data for niche modelling analysis based on the known range that comprises the eastern side of the massif where the species lives associated with riparian and beech forests (*Amat & Montori, in press*). Permits for fieldwork were granted by: Diputació de Barcelona (DIBA20140609, Servei de Fauna de la Generalitat de Catalunya: SF/0539/2019).

## Onset breeding period

In 2009, and from 2014 to 2022, we recorded the day when the *R. temporaria's* onset breeding period began in Santa Fe del Montseny (Barcelona). This area located at 1,100 m.a.s.l. is the main breeding area in the Montseny massif. During the week prior to the start of reproduction and until the day of the onset breeding, the known breeding points in the Santa Fe Valley were surveyed. The day of year (DOY) on which the males flocked to the ponds and the first clutches were detected, was considered the day of the breeding onset. The presence of isolated males was not considered indicative of the day of breeding onset. Other discontinuous data of reproductive activity based on the observation of clutches or amplexus were obtained from reports, scientific publications, technicians and researchers in the Montseny Natural Park (*Balcells, 1957*; *Pascual & Montori, 1982*; *Montori & Pascual, 1987*; *Campeny, 2001*; *Roig & Amat, 2002*; http://ornitho.cat; https://siare.herpetologica.es Martí Boada, personal communication (March, 2022) and unpublished data of the authors [1987-2004]). However, these data were not included in the analyses of the climatic factors influencing the start of the breeding period because they do not reference the day when breeding activity began, but rather, the day in which reproduction was observed. They are only considered to describe the amplitude of the breeding period in Santa Fe del Montseny.

From 2009 to 2021, daily meteorological data were obtained from "Viladrau Ws" Meteorological Station, located in Viladrau (Osona, Barcelona, at 953 m.a.s.l.; 41.84008N; 2.41877E). Data over 4 weeks (28 days) prior to the spawning were taken in order to test how previous meteorological variables influence the onset of the breeding period (Table S1). We chose 4 weeks because published studies reference the importance of the 1 to 3 week period prior to the onset of spawning (*Bea, Rodríguez-Teijeiro & Jover, 1986*; *Tryjanowski, Mariusz & Sparks, 2003*; *Loman, 2016*; *Bison et al., 2021*). Nine meteorological variables were obtained: temperature (minimal, maximal and average in °C, relative humidity in %, rain in mm, maximum speed wind (in km/h)), average of atmospheric pressure in hP, number of days with temperature below zero <0 °C and <1 °C, and solar irradiation in MJ/m$^2$ (Servei Meteorologic de Catalunya; https://meteo.cat).

To test the impact of meteorological variables on the onset of the spawning period, a 4-week period prior to the start of the spawning was compared separately with ANOVA Kruskal-Wallis test using the period (week) as a grouping variable. These four periods: 1, 2, 3 and 4 correspond to 1–7, 8–14, 9–21 and 22–28 days before the start of reproduction. A non-parametric test was used in the univariate analysis since some variables did not fulfil the condition of normality (Kolmogorov-Smirnov test).

To observe if the onset of common frog spawning activity occurs early in recent years, we performed an ordinary least - squared linear regression between the day of the onset of reproduction (DOY: day of year) and the year. To determine which meteorological variables influence the onset of egg-spawning period Factorial Analysis, using Principal Components as an extraction method. All statistical analyses were carried out with Statistica 8.0 Software. The variables used in Factorial Analysis analyses are FD4, FD3, FD2, FD1 (days with temperature below zero degrees 4, 3, 2 or 1 week before the onset of the spawning period), TX4, TX3, TX2 (temperature average 4, 3 and 2 weeks before the onset of the spawning period) and Tm4, Tm3, Tm2, Tm1 (minimum temperature 4, 3, 2 and 1 week before the onset of the spawning period). Pluviometry, maximum temperature and the temperature average in the first week prior to the onset of spawning has been not considered in the Factorial Analyses due to low contribution to explained variance in a previous exploratory analysis.

## Future distribution range contraction

We collected four-hundred georeferenced observations of *R. temporaria* at high resolution in the northeastern Iberian range of the species (Fig. 1). Nearly forty percent of localities were used originated from the Montseny massif and were recorded by us for 12 years during which we monitored amphibian populations in the natural park. We gathered nineteen bioclimatic variables at 30 s resolution from the World Climate Database (*Hijmans et al., 2005*; https://www.worldclim.org/data/cmip6/) of current and future climatic conditions under the coupled model intercomparison project phase 6 (CMIP6). We used the CNRM-CM6-1 model for a short (2021–2040) and long term (2081–2100), based on the lowest (126) and highest (585), emission scenarios to estimate the species habitat suitability. We selected maximum entropy approach (*Phillips, Anderson & Schapire, 2006*) to model habitat suitability using MaxEnt 3.4.1. because this method only uses presence data, performing well even with low sample sizes and providing reasonably accurate and interpretable models with good predictive power (*Phillips & Dudík, 2008*). We clipped raster layers to the same geographic extent covering the south westernmost margin of the Iberian Peninsula, including the Montseny massif, the Transversal Catalan Range and the Catalan Prepyrenean and Eastern Pyrenees. In order to minimise collinearity among bioclimatic variables, we used R usdm 1.1–8, to calculate the variance inflation factor (VIF) and removed highly correlated variables by performing a stepwise procedure. We checked the model performance using the size of the area under the receiver operating curve (AUC) which indicated high model suitability when it exceeded a threshold value of 0.75 (*Pearce & Ferrier, 2000*). We performed fifty replicates training the model using 25% of the presence data and quantifying variable importance by using jack-knife. Predicted species geographic distribution based on climatic suitability under current and future scenarios was represented by means of using the logistic representation of probability of occurrence.

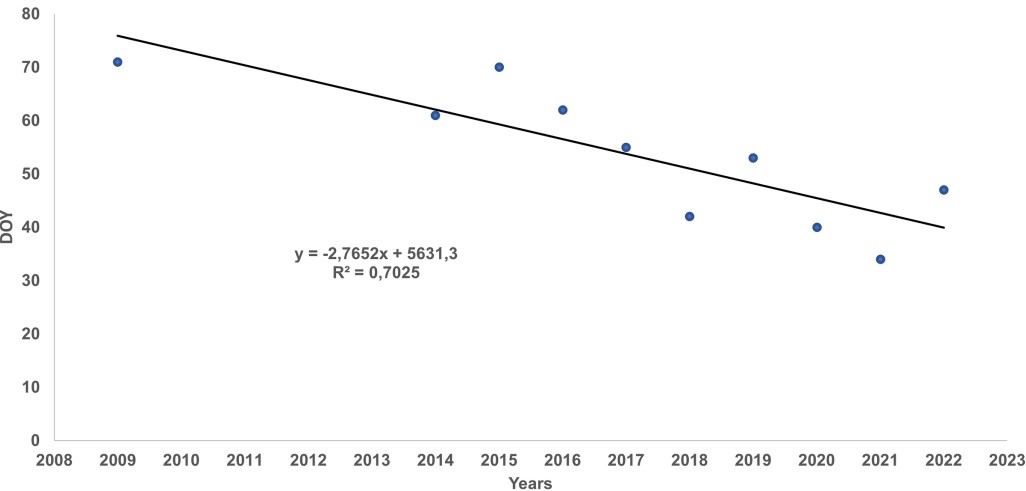

**Figure 2** Linear regression between the day of onset reproduction from January, 1 (DOY) of *Rana temporaria* in the study site and the year, for the period 2009–2022.

## RESULTS

The data on the spawning data over multiple years has been obtained from 1957 to 2022 from reports, databases and personal observations (Table S2). This table summarizes the earliest and latest data of registered spawning for each year. During this period, the earliest amplexus was registered on February 5th, in 2021 and the latest amplexus was registered on March 27th, 1984 (*Campeny, 2001*). Back then, the reproductive period of *R. temporaria* in Sta. Fe del Montseny was between early February until the second half of March. However, some sporadic spawning could occur on April (A, Montori, 1979, personal observation).

### Onset breeding period

When analysed with a linear regression on the DOY of onset spawning data recorded for the 2009 to 2022 period, we found a pattern of advancement in the beginning of reproduction (Fig. 2). The onset of reproduction, quantified as day of year (DOY), was inversely correlated with the year. The linear model obtained was significant (R = −0.7903; P = 0.0065; slope = −2.5959; SE = 2.4974) and explained 62.45% of the total variation, reflecting that part of the fluctuation in the reproductive onset has advanced over time in the last years. Between 2009 to 2022 the average for an advancement of the onset of reproduction was 26 days/decade on average in the last 14 years (Fig. 2). From 1997 to 2021 the average maximal temperature has risen 0.404 °C/decade (Fig. S1 and Table S3). This value is higher than the values published by *Peñuelas & Boada (2003)* and *Minuartia Estudis Ambentals (2016)* for the Montseny massif.

The results obtained in the Kruskal-Wallis test between periods (weeks) prior to the onset of breeding activity from 2009 to 2022 related to meteorological variables analysed are shown in Table 1. Only temperature, solar irradiation and the number of days with

Table 1 **Kruskal-Wallis analyses results.** 1st, 2nd, 3rd and 4th previous weeks to onset breeding period. Only significant variables are shown (*). Pluviometry (rain), relative humidity, wind and atmospheric pressure are not significant. Tª Av, Max and Min: Weakly average, maximum and minimum temperatures respectively. FD < 0 °C: days with temperatures below 0 °C. IR: solar radiation index. Cross analyses between weeks are shown (z statistic).

| Kruskal-Wallis test: H (3, N = 36) | | | Multiple comparisons z′ values between weeks | | | | | | |
| | | | Week | 2 | | 3 | | 4 | |
| | H | P | | z | p | z | p | z | p |
|---|---|---|---|---|---|---|---|---|---|
| Tª Av. | 11.4164 | 0.0097* | 1 | 0.9396 | 1.0000 | 2.2595 | 0.1431 | 3.1096 | 0.0112* |
| Tª Max. | 10.2592 | 0.0165* | 1 | 0.8949 | 1.0000 | 2.0806 | 0.2248 | 2.9755 | 0.0176* |
| Tª Min. | 9.57457 | 0.0226* | 1 | 1.2752 | 1.0000 | 2.1924 | 0.1701 | 2.9307 | 0.0203* |
| FD < 0 °C | 8.97857 | 0.0296* | 1 | 1.0850 | 1.0000 | 1.8569 | 0.3800 | 2.8300 | 0.0280* |
| IR | 8.38596 | 0.0387* | 1 | 1.7226 | 0.5097 | 2.0694 | 0.2311 | 2.7853 | 0.0321* |

temperature below zero degrees were significant. However, after correcting for multiple comparisons indicated that significant differences only exist between the first and fourth week. Rainfall did not seem to be a determining factor (Figs. S2 and S3). In fact, average minimum temperature for the previous week of onset spawning was 2.09 °C (STD = 0.88 °C; max = 4.07 °C; min = 0.9 °C—see meteorological data in Table S1).

The same results showed the daily variation of the minimum temperature and the interannual variance average (Fig. 3). In fact, the days prior to the start of reproduction, the minimal temperature stabilised and showed less variance than in previous weeks. Figure 4 shows the relationship between the DOY of the spawning period and the last DOY with one or three consecutive days with temperatures below zero. Both correlations were significant and highlight the importance of not having frost days in the onset of reproduction.

Factorial analysis confirmed that the variables best associated with the DOY were the number of days with temperatures below zero degrees in the previous weeks. In the first factor (Table 2 and Fig. S4), the variables related to the number of days those temperatures remained below zero and temperatures were more relevant, while in the second factor, we can find the variables related to temperature and frost days in the week previous to the onset of spawning. The first two factors represented 83.70% of the variance (Table 2), so these variables can be considered as indicative of the start of reproduction.

**Future distribution range contraction**

Habitat availability for *R. temporaria* in North-eastern Iberia, after excluding the collinear variables, was determined by five variables which had Pearson paramentric correlations between 0.695 and 0.029. Specifically, the occurrence of the species was strongly restricted to temperate conditions during the warmest month, moderate precipitation seasonality and temperatures during the driest quarter. The average test AUC for the fifty replicates was 0.896 ± 0.009 (mean ± standard deviation) indicating the model had a good fit.
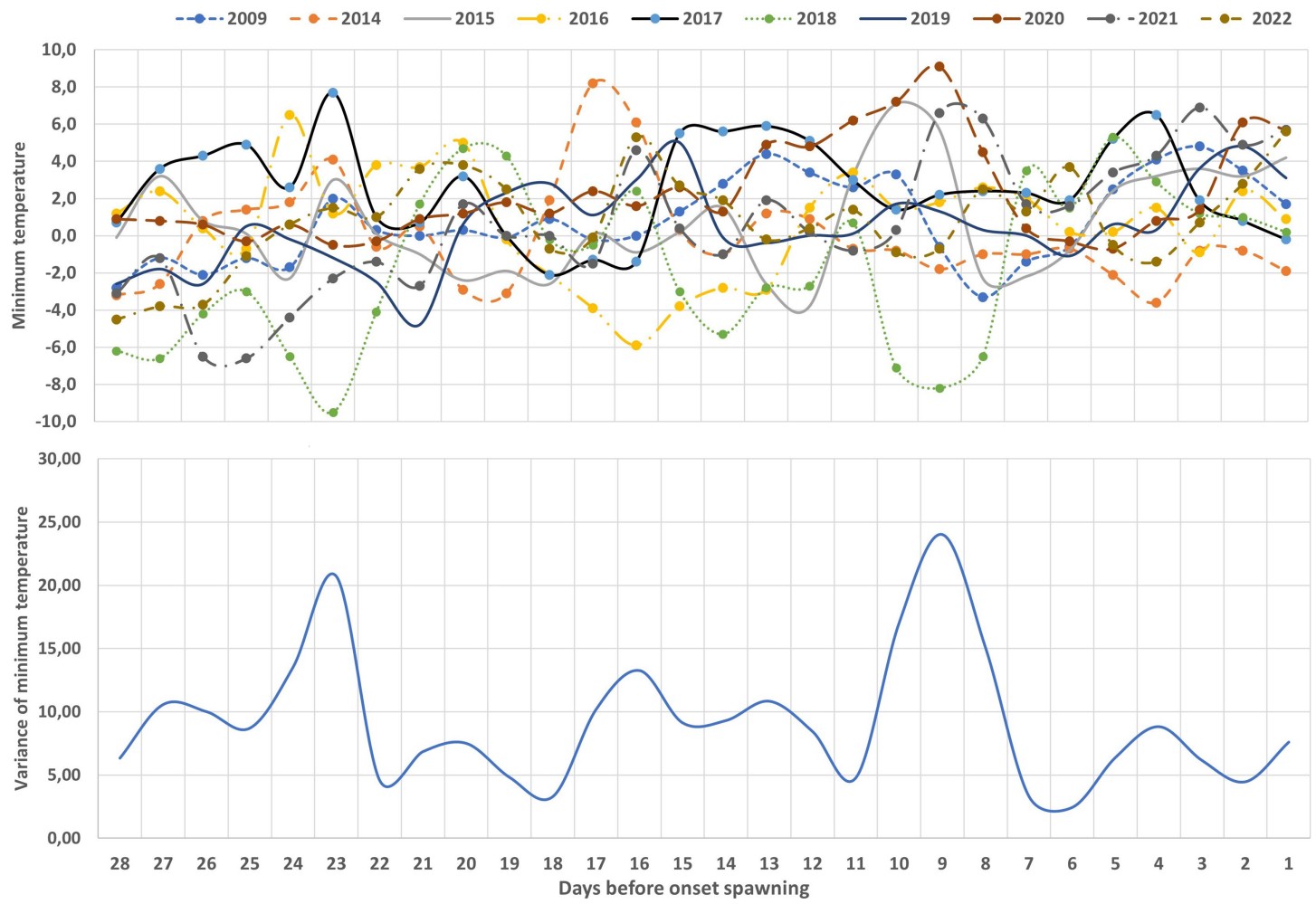

**Figure 3** Variation of temperature (°C) 4 weeks before the onset of breeding period (top) and the interannual variance of minimum temperature (°C) in the 4 weeks previous to onset of breeding period (bottom).

The predicted range of common frogs in Northeastern Iberian Peninsula was influenced by the relief as revealed by high probabilities of occurrence in the mountain areas and low probabilities at lower altitudes (Fig. 5). This led to the isolation of the populations located in the southern edge of the range, especially between the Montseny and Guilleries, and between the latter and the rest of the Catalan Transversal Range. We predicted the presence of the species in the montane ranges of the Montseny except for the massif's highest peaks. After controlling for redundancy, representing habitat suitability, the analysis retained four-six variables (parametric correlations lower than 0.7). Models had AUC tests comprised between 0.867 and 0.887, indicating good performance.

The Jackknife test revealed that mean temperatures in the wettest and driest quarter, maximum temperatures in the warmest month, precipitation in the warmest quarter and precipitation seasonality were the most influential variables, followed by precipitation and temperature seasonality and isothermality. Models based on scenarios ssp 128 and ssp 585
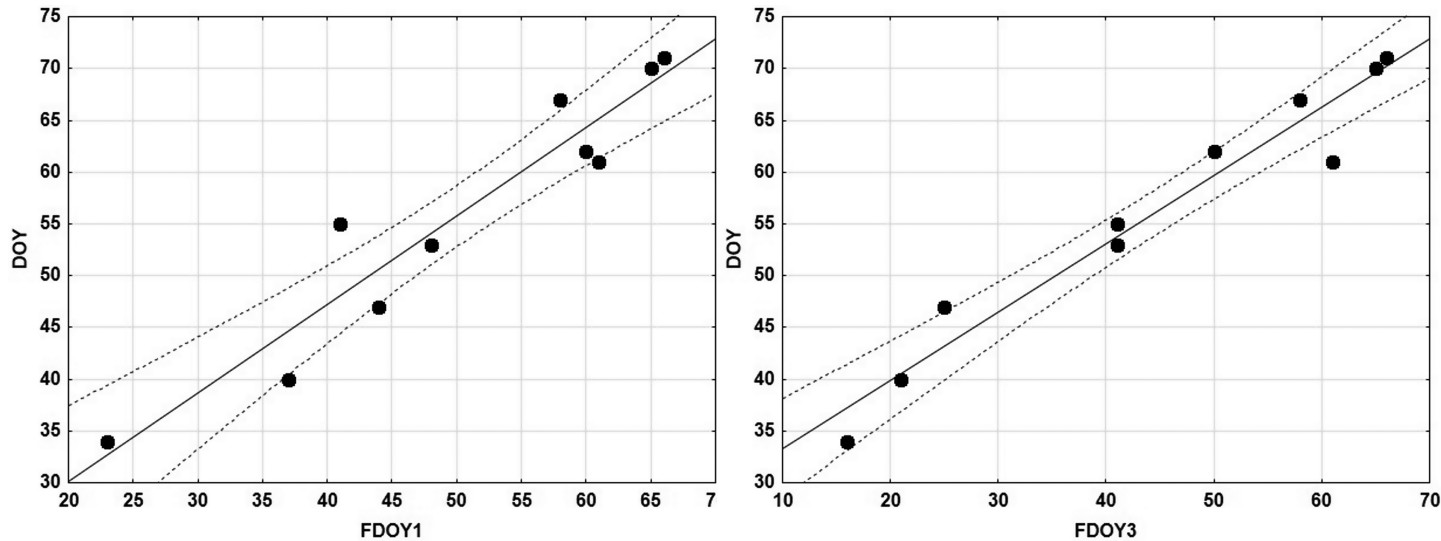

**Figure 4** Relationship between the last day of year with air temperature below 0 °C (FDOY1) previous to onset spawn and the DOY of onset egg-spawning of the common frog *Rana temporaria* (Left), and between the last day of year with three consecutive days with air temperature below 0 °C (FDOY3) and the DOY of onset egg-spawning (Right). Each dot corresponds to one year. DOY "day of the year".

**Table 2 Factor coordinates of the variables, based on correlations and eigenvalues of correlation matrix and related statistics.** FD: Number of days with temperature below 0 °C. 1, 2 3, and 4: weeks before onset breeding. TX, maximum temperature; Tm, minimum temperature. Bold scores: significant variables in each factor.

| Variable | Factor 1 | Factor 2 |
|---|---|---|
| **DOY** | **0,652040** | 0,004689 |
| **FD4** | **−0,874749** | 0,132583 |
| **FD3** | **−0,859343** | −0,363795 |
| **FD2** | **0,843404** | −0,443237 |
| **FD1** | −0,453862 | **−0,836335** |
| **Tm1** | 0,640920 | **0,714930** |
| **Tm2** | **0,963254** | −0,122021 |
| **Tm3** | **0,859055** | −0,358827 |
| **TX3** | **0,859055** | −0,358827 |
| **TX2** | **0,924646** | −0,232611 |
| **TX4** | 0,684461 | −0,639220 |
| **TX1** | −0,561564 | −0,553076 |
| **Tm4** | **0,851790** | −0,464659 |
| **Eigenvalue** | 8.115321 | 2,765514 |
| **% total Variance** | 62.42555 | 21.27329 |
| **Cum. Eigenvalue** | 8.115321 | 10.88084 |
| **% Cum. Variance** | 62.42555 | 86.69873 |

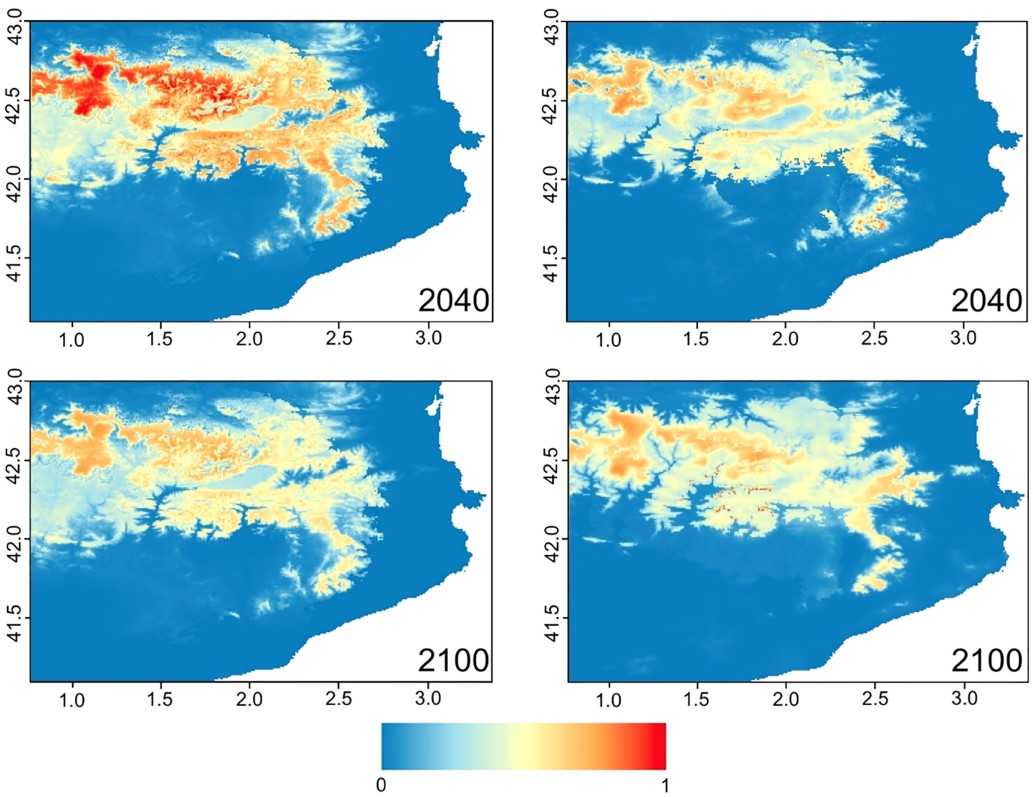

**Figure 5** Habitat availability and categorical predictability for *Rana temporaria* in northeastern Iberia during the periods 2021–2040 and 2081–2100 predicted by MaxEnt algorithm based on CNRM-CM6 1 under low (left) and high (right) emission scenarios.

(Fig. 5) considerably reduced the probability of species occurrence when comparing the predictions obtained between 2040 and 2100. Our results highlighted the isolation between the populations in the southernmost edge, and between them and the Pyrenean ones.

In the Montseny massif, the future predicted ranges in 2040 were similar that currently occupied by the species, but occurrence probability drops and rejected an altitudinal shift towards the highest top mountains. The worst scenario, high level of emissions in 2100, predicts species occurrence with low probability and only in the eastern side of the massif, suggesting the failure of its persistence in Montseny. Thus, in all our predicted scenarios, low and high emissions entailed a dramatic reduction in habitat availility in the Montseny massif within this century (Fig. S5).

## DISCUSSION

### Onset breeding period

The common frog in the Montseny massif had advanced its spawning onset following the increase in temperatures due to global warming. Significantly, the four earliest dates of onset spawning periods were in 2019, 2020, 2021 and 2022 (Fig. 2). Between 2009 and 2022 the average of advancement of the onset of reproduction is 26 days/decade (Fig. 2). Our estimated advancement of the onset of the breeding season in the Montseny massif is

high, but included within the range found by *While & Uller (2014)* for *R. temporaria* (−28.1 days/decade).

Probably, the heavy Mediterranean climatic impact on the Montseny massif. The southern distribution limit of many eurosiberian species, associated with the rapid increase in temperatures during recent years (*Minuartia Estudis Ambentals, 2016*), could be the causes behind this great advancement. Supporting this data (*Minuartia Estudis Ambentals, 2016*), estimates an increase in the average annual temperature of 0.7 °C for the decade (2012–2021) in the Coastal/Pre-coastal Region (Montnegre-Corredor Natural Park), to which the Montseny massif is close.

This earliest onset reproduction date in amphibians has been pointed out by different authors. *Beebee (1995)*, detected an advancement in the beginning of reproduction of some species, which was linked to global warming. Multispecies meta-analyses have suggested that amphibians are the taxa showing the strongest phenological advancement in response to global warming, with an average breeding advancement ranging from 2.6 days per decade (*Parmesan, 2007*) to 6.1 days per decade (*While & Uller, 2014*). The mean advancement in *R. temporaria*'s onset of spawning found by *While & Uller (2014)* was −5.95 day/decade (SD = 11.92; maximum advancement; −28.1 day/decade). *While & Uller (2014)* have clearly shown that amphibian populations consistently anticipate their breeding periods, determined by trends of global warming, particularly at high latitudes (*Terhivuo, 1988*).

Our results are in accordance with *Ficetola & Maiorano (2016)* who conclude that temperature increase was the major factor behind phenological advancement, while the impact of precipitation on phenology was weak. In fact, we didn't obtain any significant relation between rain and DOY of onset breeding (Table S1 and Figs. S2 and S3).

For the end of the last century, other authors (*Peñuelas & Boada, 2003*; *Minuartia Estudis Ambentals, 2016*), estimated an increase in air temperatures ranging from 0.24 to 0.3 °C per decade in the Montseny massif. However, the data obtained in this study from 1996 to 2021 in the same weather station indicate that this increase has accelerated to 0.404 °C/decade on average of maximal temperature (Fig. S1), according to the observations by *Peñuelas & Boada (2003)* who consider that the increase in temperatures has accelerated over recent decades.

This advancement demonstrates a great plasticity in spawning dates in relation to climate change. However, *Phillimore et al. (2010)* in Britain observed that although all populations exhibit a plastic response to temperature, spawning earlier in warmer years, differences within populations in their first spawning dates are dominated by local adaptation. Climate change projections for Britain in 2050–2070, will require an advancement of between 21 and 39 days for the first spawning date, but *Phillimore et al. (2010)* indicate that plasticity alone will only enable an advancement of 5 to 9 days, and need a microevolutionary and gene flow challenge for an earlier first spawning date of between 16–30 days over the next 50 years. However, our data demonstrates that from 2009 to 2022, the advance of the onset of spawning has been 26 days, more than the data shown by the other authors. Perhaps living and surviving in the edge has been favourable for more plastic individuals due to the suboptimal and higher variability of environmental

conditions. According to this hypothesis, *Lesbarrères et al. (2007)* found that metamorphic size was positively correlated with individual heterozygosity and fitness in *R. temporaria*'s Nordic distribution limit. These results suggest that genetic variability may be an important component for individual fitness in the common frog (*Lesbarrères et al., 2007*). *Ruthsatz et al. (2022)* found that *R. temporaria* has a high acclimation capacity in the temperate populations, which could reflect an adaptation to increased thermal variability to enable survival in heterogeneous or suboptimal environments, especially during the larval stage. Generally, wide-ranging species, such as *R. temporaria*, exhibit a greater capacity for thermal acclimation of upper thermal tolerance than narrowing-ranging species, suggesting that selection for acclimation ability may have been a key factor that enabled geographic expansion into areas with greater thermal variability (*Markle & Kozak, 2018*). These authors demonstrate that young larvae of *R. temporaria* are the most temperature-sensitive life stage and highlights that early life stages might define the climate change vulnerability of amphibian populations. Therefore, this widely distributed species possesses a high capacity for developmental, morphological, and physiological plasticity. In this sense, in the Montseny massif, *R. temporaria* has chosen, throughout the years, the week with positive minimal temperature to onset spawn (Average minimal $T^a$ for the previous week of onset spawning = 2.09 °C; STD = 0.88 °C; Max = 4.07; Min = 0.9), with a great variability in DOY between years (Fig. 2) but low variability in spawning environmental conditions.

*Enriquez-Urzelai et al. (2018)* indicates that *R. temporaria* tadpoles are an eurythermal organism and thermal niches vary across latitudinal and altitudinal clines. Furthermore, at both extremes of the latitudinal gradient, plasticity is maximal. *Enriquez-Urzelai et al. (2020)* indicate that mixed signals of niche evolution and conservation at the species level and generalized conservation in thermal niches at the population level indicate that the pace of niche evolution in *R. temporaria* might be too slow to compensate climate change (mainly among terrestrial life cycle stages) according to *Phillimore et al. (2010)*.

Previous studies identified a strong relationship between snowmelt dates and spawning dates (*Terhivuo, 1988*; *Corn & Muths, 2002*; *Corn, 2003*; *Bison et al., 2021*), thereby confirming a high degree of plasticity in terms of breeding timing (*Muir, Biek & Mable, 2014*). *Bison et al. (2021)* observe in *R. temporaria* a relationship with the DOY of snowmelt and DOY of middle spawning activity. In our case, the location of the Montseny massif in the southern limit of distribution where the climatic conditions are Mediterranean, makes it impossible to determine the snowmelt day because the snow is not present every year and the pluviometry in the first month of the year is very scarce. Consequently, to compare the DOY of the onset of snowmelt with the theoretical snowmelt, we used the DOY of the last 1 or 3 days when there were temperatures below zero. Our results confirm one highly and significant relation between the DOY of spawning and DOY of the last frost days (Fig. 4) according to the results obtained by *Bison et al. (2021)*.

*Enriquez-Urzelai et al. (2020)* indicate that high-elevation populations had slightly wider tolerance ranges driven by increases in heat tolerance but a lower potential for acclimation. Plausibly, wider thermal fluctuations at high elevations could be extrapolated

to distribution limits like the Montseny populations. Then, extreme events could have more unpredictable effects on populations living in the edge of their distribution, as in the case of Montseny populations. Increases in air temperatures and more frequent extreme events (*e.g.*, heat waves) due to climate warming will produce changes in pond temperatures (*Enriquez-Urzelai et al., 2019b*). Species could respond *in situ* by increasing upper thermal limits through genetic adaptation or phenotypic plasticity. However, the potential for adaptation and plastic responses to shift thermal tolerances seems limited in ectotherms, relative to predicted environmental changes (*Enriquez-Urzelai et al., 2018*, *2019b*). Thus, species or populations living closer to their maximum thermal tolerance (*e.g.*, southernmost populations of *R. temporaria*) might be especially vulnerable to climate change (*Duarte et al., 2012*; *Gerick et al., 2014*).

Furthermore, although the *R. temporaria* adults show great adaptability or plasticity in terms of the onset of the spawning period, the vulnerability of the larvae and adults when faced with the increase in extreme phenomena in a scenario of climate change could be high. *Montori et al. (2011)* observe that mortality of amphibians increases due to more frequent sudden drops in temperature, which can affect both larvae and adults (Fig. S6). If the advancement of reproduction continues in the coming years, the probability of post-spawning frost days will increase, given the variability of the climate in the context of an increase in extreme phenomena. Reinforcing this hypothesis, *Bison et al. (2021)* observed that the number of frost days during egg-development increased more quickly at high elevation. Then, early spawn could lead to an additional rise in larval or spawn mortality due to the greater number of frost days (*Montori et al., 2011*). *Enriquez-Urzelai et al. (2019b)* predicted extensive decreases in climatic suitability in Southern Europe, which harbors a significant fraction of the species' genetic diversity. Moreover, these authors suggest that *R. temporaria*'s climatic suitability decreases at locations where water is scarce, or ponds reach temperatures of >40 °C. This temperature is close to the critical thermal maxima of *R. temporaria* tadpoles (*Enriquez-Urzelai et al., 2019b*). In the shallow ponds in which *R. temporaria* commonly breeds, thermal conditions depend on the degree of insolation and depth. Tadpoles suffer under great thermal fluctuations in ponds, and thus extreme pond temperatures may set the limits of the distribution of pond-breeding amphibians. The reproduction ponds studied by *Campeny (2001)* in Santa Fe del Montseny show a maximum average annual temperature between 20.83 °C and 26.00 °C, with maximums in summer of over 32 °C. In addition, these ponds are very shallow and the water has average depths oscillating from 8.19 cm with SD of 4.54 cm, to 15.32 cm with SD of 8.19 cm. In these abiotic conditions, the rising temperatures could increase the risk of larval mortality in the future.

## Future distribution range contraction

The Montseny massif and the neighbouring Montnegre massif are the southernmost edge in East Iberia of ectothermic vertebrates adapted to temperate conditions (*i.e.*, *Lacerta bilineata*, *Podarcis muralis*, *Vipera aspis* and *Zamenis longissimus*) that will likely be affected by the rise in temperatures and dryness. In this context the southwesternmost

populations of common frog are an even more interesting species model because of their cold adaptation capabilities.

Although our results provided evidence of range contraction in the Montseny (Fig. 5 and Fig. S5), they failed to confidently predict the local extinction of the species. Amphibians are mostly low vagile and small vertebrates which rely on microclimatic conditions (*Smith & Green, 2005*; *Hoffmann, Cavanough & Mitchell, 2021*), far from the scope of the climatic niche analysis. Range contraction is frequently predicted by climatic niche modelling in analysis of the consequences of global warming in cold-adapted species (*Pauli et al., 2012*; *Ernakovich et al., 2014*). In the Montseny massif, most of the reproductive ponds are placed in the east side, more temperate and humid, whereas the species experiences the harshest conditions in the southwestern slopes exposed to more Mediterranean climate. In this area, only four reproductive sites are known and the estimated number of reproductive females is dramatically small and undoubtedly, they are the most endangered and could disappear during the coming decades.

Common frogs exhibit a large ecological plasticity, living in mashes at the sea level and high altitude lakes in the Pyrenees (*Kuzmin et al., 2009*). In the Montseny massif they are associated with temporal pools, in most cases in open areas within the margins of the riparian forest (F. Amat, 2021, unpublished data.). The rise in temperatures increases the risk of pond desiccation leading to the failure of the reproduction because most of them depend on irrigation from neighbouring streams to maintain a long hydroperiod. Early reproduction, as our results indicated, may compensate for the shortening of the pond's hydroperiod. However, even then, it cannot outweigh extreme spring dryness that predictably could become more frequent over time in the Mediterranean region (*Quintana-Seguí et al., 2016*). This phenomenon could also play a negative role in the preservation of the riparian forest that provides humid refugia for the activity of adults after reproduction.

Air temperatures fluctuate more than water temperatures (*Feder & Hofmann, 1999*). Thus, terrestrial stages are more likely to encounter more extreme temperatures than aquatic larvae. In contrast, they can use behavioural thermoregulation. This may not only allow individuals to escape unwanted temperatures (*Kearney & Porter, 2009*), but may also weaken directional selection based on thermal traits, through a process known as the 'Bogert effect' (*Huey et al., 2009*; *Buckley, Ehrenberger & Angilletta, 2015*). Oppositely, in water, the potential of behavioural thermoregulation to buffer heat or cold waves may be more limited.

In amphibians, metamorphic and juvenile stages can thermoregulate better than larval stages. Although most forecasts with regard to the consequences of global warming for biodiversity conservation ignore how thermal tolerance varies between life stages, the viability of the weakest link could restrict the future distribution of a species (*Briscoe et al., 2012*; *Pincebourde & Casas, 2015*). *Enriquez-Urzelai et al. (2020)* demonstrate that larval acclimation to high temperatures increases larval heat tolerance, but may reduce cold tolerance in immediately subsequent stages. If sudden drops in temperature are more frequent due to an advance in the reproduction date, this can endanger the survival of some tadpole pond populations.

It is likely that wider thermal fluctuations at high elevations favour more tolerant but less plastic phenotypes, thus reducing the risk of encountering stressful temperatures during unpredictable extreme events (*Enriquez-Urzelai et al., 2020*). This wider thermal fluctuation also occurs in the Montseny massif population due its location in a Mediterranean climate, but we can't conclude if the Montseny population presents more adaptability or plasticity, and as other authors suggest, if this adaptability has a genetic basis (*Phillimore et al., 2010*). Future studies should address this problem in populations that survive in the southernmost limit of its wide distribution.

## ACKNOWLEDGEMENTS

We want to thank all those people who provided us with data on the phenology of *R. temporaria* and especially Joan Manel Roig Fernández for providing data on species phenology prior to the implementation of the current project, to Marc Franch and Adrià Jordà for assisting in some field research in Santa Fe and to Martí Boada, a pioneer in the annotation of phenological data from Montseny. This field data book undoubtedly has an invaluable historic utility. We also want to thank the use of data from the databases of the Institut Català d'Ornitologia (ICO: http://ornitho.cat) and the Asociación Herpetológica Española (AHE; https://siare.herpetologica.es) and Societat Catalana d'Herpetologia (SCH: https://soccatherp.org/).

### Funding

The Diputació de Barcelona-Parc Natural del Montseny provided financial support to the field surveys (reference number 2015/3456). There was no additional external funding received for this study. The funders had no role in study design, data collection and analysis, decision to publish, or preparation of the manuscript.

### Grant Disclosures

The following grant information was disclosed by the authors:
The Diputació de Barcelona-Parc Natural del Montseny: 2015/3456.

### Competing Interests

The authors declare that they have no competing interests.

### Author Contributions

- Albert Montori conceived and designed the experiments, performed the experiments, analyzed the data, prepared figures and/or tables, authored or reviewed drafts of the article, and approved the final draft.
- Fèlix Amat conceived and designed the experiments, performed the experiments, analyzed the data, prepared figures and/or tables, authored or reviewed drafts of the article, and approved the final draft.
## Animal Ethics

The following information was supplied relating to ethical approvals (*i.e.*, approving body and any reference numbers):

Permission for manipulation animals was given by the experimental committee of Generalitat de Catalunya (Spain).

## Field Study Permissions

The following information was supplied relating to field study approvals (*i.e.*, approving body and any reference numbers):

Permits for fieldwork were granted by: Diputació de Barcelona (DIBA20140609, Servei de Fauna de la Generalitat de Catalunya: SF/0539/2019).

## Data Availability

The raw data is available in the Supplemental Files and at Zenodo:

-Montori, Albert, & Amat, Fèlix. (2022). Meteorlogial Data from Viladrau WS Meteorological Station. https://meteo.cat [Data set]. Zenodo. https://doi.org/10.5281/zenodo.7186463.

-Montori, Albert, & Amat, Fèlix. (2022). Daily meteorological data from January 2005 to February 18, 2022 from Viladrau Meteorological Station (Catalonia, Spain) [Data set]. Zenodo. https://doi.org/10.5281/zenodo.7186516.

-Montori, Albert, & Amat, Fèlix. (2022). Used Coordinates of *Rana temporaria* from Spain. [Data set]. Zenodo. https://doi.org/10.5281/zenodo.7186550.

The data used in this work can also be requested through this form: https://www.meteo.cat/wpweb/serveis/formularis/peticio-dinformes-i-dades-meteorologiques/peticio-de-dades-meteorologiques/. The relevant parameters were: Locality: Viladrau. (Catalonia, Spain); Meteorological station: Viladrau WS; Coordinates: UTM X: 451743 Y: 4632184; Altitude: 953 m.

## Supplemental Information

Supplemental information for this article can be found online at http://dx.doi.org/10.7717/peerj.14527#supplemental-information.

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
