# Peer review of "Surviving on the edge: present and future effects of climate warming on the common frog (Rana temporaria) population in the Montseny massif (NE Iberia)"

_PeerJ, doi:10.7717/peerj.14527_

## Round 0.1 · original submission · Major Revisions

The ms reports interesting data on the effect of climate change on an isolated group of common frog population at the edge of their distribution range. Please address the suggested changes and resubmit.

Reviewer 1 ·

Basic reporting

In this article, Montori and Amat investigated the impact of climate change on spawn period of common frogs (Rana temporaria) in Montseny massif on three levels - 1) is there an advance of onset of spawn period due to warming temperature? 2) the influence of climatic variables. And 3) prediction of population dynamics. Basically, I think the logic of this paper is clear, and the authors have made important contribution to this research field, which would help understand the impact of climate change on the life history of animals, especially for amphibians. However, the English writing of this paper needs to be substantially improved. There are still a lot of grammar mistakes and typos.

Experimental design

I think the research meets the Aims and Scope of the journal well, which was well defined and provided information on how climate change affects the life history of amphibians.The technical & ethical standards are enough. Methods described with sufficient details.

Validity of the findings

The data have been provided, which are robust and statistically sound. Conclusion are well stated.

Additional comments

Please note that the Figure 10 and Figure 11 are actually tables, which should be corrected. I think the Figure 2, 4, 8 and 9 do not provide enough information, so that I suggest to move them to supplementary materials.

·

Basic reporting

The ms reports interesting data on the effect of climate change on an isolated group of common frog population at the edge of their distribution range. Unfortunately, the English wording of the ms must be improved considerably (I provide some suggestions in the commented ms, but a fluent speaker should go over the whole ms), there are many phrases, which are hardly understandable. Moreover, the ms is not concise enough, the reader is often lost in details, which are not really neccessary to report.

References are well chosen (but with some errors in the list, see commented pdf) and the article structure is adequate. Conclusion follow from the data reported.

Experimental design

This is not an experimental study. Data are observational. Analyses are adequate, but there are some which could be excluded without loss of information (see commented pdf).. The research presented falls into the scope of the journal.

Validity of the findings

This is another proof for the impact of global warning on the reproduction of amphibian. While evidence is increasing and available for several species, this study adds clearly to the knowledge because the prediction of effects until the end of this century demonstrate that common frogs from Montseny are prone to extinction.

Additional comments

My recommendations and suggestions are written in detail in the attached commented ms pdf.

·

Basic reporting

The use of English language needs to be substantially improved. Below are some examples, but the entire manuscript needs to be carefully read through for many instances of language correction:

line 23: change to "this species"

line 25-26: this is difficult to understand - is this meant to be something like "more than the 0,1ºC indicated by estimates for the second half of previous century"

line 67: change to "finding"

line 74: change to "shift of their ranges"

line 74: change to "Changes of geographic". Throughout the manuscript please look for more instances where "on" is used but its meant to be "of".

line 83: change "your" to "their".

line 86: correct decimal point here.

line 113: change to "in recent years". on the next line, change to "faces".

Experimental design

Some details of the manuscript were difficult to follow due to the need for language improvements, so some more changes needed to aspects such as experimental design and validity of findings may surface after the editing for language. Reading through this version, the main comment I had was about the need for some more details in the methods. Specifically, around line 134, please could you provide some more details about the methods for recording the day of the onset of the breeding period. Was this only recorded at one location or area? Is it known that all the locations have breeding onset at the same day? What observation was made to confirm the breeding period had begun? I guess daily monitoring effort must have been made leading up to the day of breeding onset, if so please describe this, including if and how the monitoring was standardised or done in some systematic way.

Validity of the findings

no comment

Additional comments

line 443-464: possibly this conclusion section is not needed? Lines 439-441 discuss the possibility of future research, which might be a good point to end on. Definitely, lines 460-464 are an almost verbatim repeat of the lines 436-441, which should not be done. Please either delete the conclusions or re-write it to summarize better the results in ways that have not already been done elsewhere in the manuscript.

line 252: change this to "Figure 7".

line 287: change the citation here to have normal formatting.

---

## Round 0.2 · Minor Revisions

The revised manuscript is fine except for very minor typographical errors.

·

Basic reporting

the revised ms is fine except for very minor typographical errors.

Experimental design

no comment

Validity of the findings

Findings are interestings and now well-presented.

Additional comments

I found very few minor typographical errors, which should be corrected before publication (see attached file). The ms has been considerably improved by the revision. Congratulations,

Ulrich

---

## Round 0.3 · accepted · Accept

Congratulations. Thank you for your submission to PeerJ.